# Is Multiple Object Tracking a Matter of Specialization?

**Gianluca Mancusi**     **Mattia Bernardi**     **Aniello Panariello**     **Angelo Porrello**

**Rita Cucchiara**                              **Simone Calderara**

AImageLab - University of Modena and Reggio Emilia
`name.surname@unimore.it`

## Abstract

End-to-end transformer-based trackers have achieved remarkable performance on most human-related datasets. However, training these trackers in heterogeneous scenarios poses significant challenges, including negative interference – where the model learns conflicting scene-specific parameters – and limited domain generalization, which often necessitates expensive fine-tuning to adapt the models to new domains. In response to these challenges, we introduce Parameter-efficient Scenario-specific Tracking Architecture (PASTA), a novel framework that combines Parameter-Efficient Fine-Tuning (PEFT) and Modular Deep Learning (MDL). Specifically, we define key scenario attributes (*e.g.*, camera-viewpoint, lighting condition) and train specialized PEFT modules for each attribute. These expert modules are combined in parameter space, enabling systematic generalization to new domains without increasing inference time. Extensive experiments on MOT-Synth, along with zero-shot evaluations on MOT17 and PersonPath22 demonstrate that a neural tracker built from carefully selected modules surpasses its monolithic counterpart. We release models and code.

## 1 Introduction

Video Surveillance is essential for enhancing security, supporting law enforcement, improving safety, and increasing operational efficiency across various sectors. In this respect, Multiple Object Tracking (MOT) is a widely studied topic due to its inherent complexity. Nowadays, MOT is commonly tackled with two main paradigms: tracking-by-detection (TbD) [3, 51, 61, 28, 43, 38, 29] or query-based tracking [56, 58, 63, 12] (*i.e.*, tracking-by-attention). Although tracking-by-detection methods have proven effective across multiple datasets, their performance struggles to scale on larger datasets due to the non-differentiable mechanism used for linking new detections to existing tracks. To this end, query-based methods are being employed to unify the detection and association phase.

Nevertheless, training such end-to-end transformer-based methods presents significant challenges, as they tend to overfit specific scenario settings [37, 55] (*e.g.*, camera viewpoint, indoor *vs*. outdoor environments), require vast amounts of data [63], and incur substantial computational costs. Moreover, these methods degrade under domain shifts, struggling to outperform traditional TbD methods.

In light of these challenges, we propose a novel framework, Parameter-efficient Scenario-specific Tracking Architecture (PASTA), aimed at reducing the computational costs and enhancing the transfer capabilities of such models. Leveraging Parameter Efficient Fine-Tuning (PEFT) techniques [16, 34] can significantly decrease computational expenses and training time, starting with a frozen backbone pre-trained on synthetic data. However, the model may still experience *negative interference* [49,

38th Conference on Neural Information Processing Systems (NeurIPS 2024).

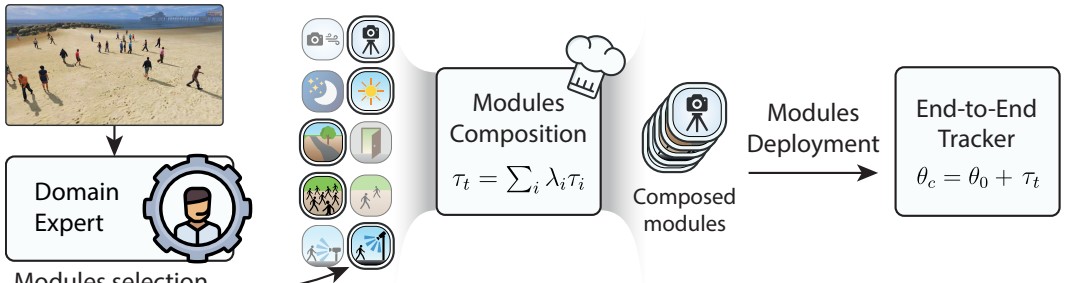

Figure 1: Given a scene, we select the modules corresponding to its attributes, such as lighting and indoor/outdoor. These modules are composed and then deployed, yielding a specialized model.

50, 37, 55], a phenomenon for which training on multiple tasks (or scenarios) causes the model to learn task-specific parameters that may conflict. For instance, if the model learns parameters tailored for an indoor sports activity, it could detrimentally affect its performance on a novel outdoor scene depicting people walking. To this end, inspired by Modular Deep Learning (MDL) [35], we employ a lightweight expert module for each attribute, learn them separately, and finally compose them efficiently [17]. This approach – depicted in Fig. 1 – is akin to a chef preparing a pasta dish. Each ingredient (*i.e.*, module) is prepared individually to preserve its unique flavor and then combined harmoniously to create a balanced dish. Moreover, as pasta must be perfectly *al dente* to serve as the ideal base for various sauces, the pre-trained backbone should be robust and well-tuned to serve as the foundation for the modules. These modules must be combined effectively to ensure the model performs well across diverse scenarios. Conversely, the result will be sub-optimal if incompatible modules are mixed – analogous to combining ingredients that do not complement each other. Indeed, combining contrasting modules can lead to ineffective handling of diverse tasks.

Notably, such a modular framework brings two advantages: it avoids negative interference and enhances generalization by leveraging domain-specific knowledge. Firstly, starting from a pre-trained backbone, we train each module independently to prevent parameter conflicts, ensuring that gradient updates are confined to the relevant module for the specific scenario. This assures that parameters learned for one attribute do not negatively impact the performance of another. Secondly, the modular approach allows us to exploit domain knowledge fully, even when encountering a novel attribute combination. Indeed, as shown in Sec. 5.5, our approach is effective even in a zero-shot setting (*i.e.*, without further fine-tuning on the target dataset). Moreover, the selection of the modules may be done automatically or in a more realistic production environment by video surveillance operators.

To evaluate our approach, we conduct extensive experiments on the synthetic MOTSynth [10] and the real-world MOT17 [8] and PersonPath22 [44] datasets. The results show that PASTA can effectively leverage the knowledge learned by the modules to improve tracking performance on both the source dataset and in zero-shot scenarios. To summarize, we highlight the following main contributions:

- We propose PASTA, a novel framework for Multiple Object Tracking built on Modular Deep Learning, enabling the fine-tuning of query-based trackers with PEFT techniques.
- By incorporating expert modules, we improve domain transfer and prevent negative interference while fine-tuning MOT models.
- Comprehensive evaluation confirms the validity of our approach and its effectiveness in zero-shot tracking scenarios.

## 2 Related works

**Multiple Object Tracking.** The most widely adopted paradigm for Multiple Object Tracking (MOT) is *tracking-by-detection* (TbD) [3, 51, 61, 28, 43, 38]. First, an object detector (*e.g.*, YOLOX [14]) localizes objects in the current frame. Next, the association step matches detections to tracks from the previous frame by solving a minimum-cost bipartite matching problem, with the association cost defined in various forms (*e.g.*, IoU [3, 61], GIoU [40], or geometrical cues [29, 33]). This pairing typically occurs immediately after propagating the previous tracks to the current frame using a motion model (*e.g.*, Kalman Filter [19]). Notably, methods following such paradigm have succeeded on

complex human-related MOT benchmarks [8, 9, 47, 44]. In TbD, the detection and data-association steps are equally crucial to accurately localizing and tracking objects. Recent works [65, 1] have attempted to unify these steps; however, progress toward a fully unified algorithm was constrained by a significant limitation – the data association process (*e.g.*, the Hungarian algorithm [20]) is inherently non-differentiable. An initial effort was made by Xu *et al.* [53] that proposed a differentiable version of the Hungarian algorithm, later advanced by end-to-end transformer-based trackers [31, 58, 63, 56, 13].

However, transformer-based trackers (also known as tracking-by-attention) require large amounts of data to achieve decent generalization capabilities [58, 31]. Due to the data scarcity in MOT, these models often overfit to the specific domain they were trained on, which hampers their ability to generalize to different domains [17, 21, 37].

**Modular Deep Learning (MDL).** Considering recent trends in the field of deep learning, state-of-the-art models have become increasingly larger. Consequentially, fine-tuning these models has become expensive; concurrently, they still struggle with tasks like symbolic reasoning and temporal understanding [35]. Recent learning paradigms based on *Modular Deep Learning* (MDL) [35] can address these challenges by disentangling core pre-training knowledge from domain-specific capabilities. By applying modularity principles, deep models can be easily edited, allowing for the seamless integration of new capabilities and the selective removal of existing ones [26, 36].

Specifically, lightweight computation functions named *modules* are employed to adapt a pre-trained neural network. To do so, several fine-tuning techniques could be used to realize these modules, such as LoRA [16], (IA)$^3$ [25], and SSF [23]. These multiple modules can be learned on different tasks such that they can specialize in different concepts [32]. At inference time, not all modules have to be active at the same time. Instead, they can be selectively utilized as needed, either based on prior knowledge of the domain or dynamically in response to the current input. To establish which modules to activate, it is common practice to rely on a *routing function*, which can be either learned or fixed. Finally, the outputs of the selected modules are combined using an *aggregation function*. To minimize inference costs, this process is usually performed in the parameter space rather than the output space, an activity often referred to as *model merging* [54]. Specifically, a single forward pass is performed using weights generated by a linear combination of those selected by the routing function.

**Domain adaptation and open-vocabulary approaches in MOT.** Currently, domain adaptation techniques have only been applied to tracking-by-detection methods, with GHOST [43] and DARTH [42] serving as notable examples. In particular, GHOST adapts the visual encoder employed to feed the appearance model by updating the sufficient statistics of the Batch Normalization layers during inference. In contrast, our approach regards tracking-by-attention approaches and adapts the entire network. Moreover, DARTH employs test-time adaptation (TTA) [24] and Knowledge Distillation, requiring multiple forward passes and entire sequences, making it computationally heavy and less practical for real-time use. In contrast, our method is entirely online and requires only basic target scene attributes, with no further training during deployment.

Recent advances in zero-shot tracking have focused on *open-vocabulary tracking*, where the model can track novel object categories by prompting it with the corresponding textual representation. In this respect, methods like OVTrack [22] and Z-GMOT [48] leverage CLIP [39] and language-based pre-training, while OVTracktor [7] extends tracking to any category. Our method does not use open-vocabulary models but emphasizes domain knowledge transfer in end-to-end trackers.

## 3 Preliminaries

**Efficient fine-tuning.** Given the substantial size of recent vision backbones, often consisting of hundreds of millions of parameters, adapting them to new scenarios is computationally expensive, both in terms of time and memory requirements. To tackle the above problems, Parameter Efficient Fine-Tuning (PEFT) started to take place in recent literature. Among these methods, Low-Rank Adaptation (LoRA) [16] excels at such purpose. Specifically, LoRA adapts a pre-trained weight matrix $\boldsymbol{\theta}_0 \in \mathbb{R}^{d \times k}$, with $d$ and $k$ being the dimensions of the matrix, by leveraging a low-rank decomposition $\boldsymbol{\theta}_0 + \Delta\theta = \boldsymbol{\theta}_0 + BA$, where $B \in \mathbb{R}^{d \times r}, A \in \mathbb{R}^{r \times k}$, and $r \ll \min(d, k)$. During training, $\boldsymbol{\theta}_0$ is kept frozen, while the smaller $A$ and $B$ matrices are instead trainable, making the process highly efficient. The forward pass becomes $h = \boldsymbol{\theta}_0 x + BAx$, where $x$ are the input features.

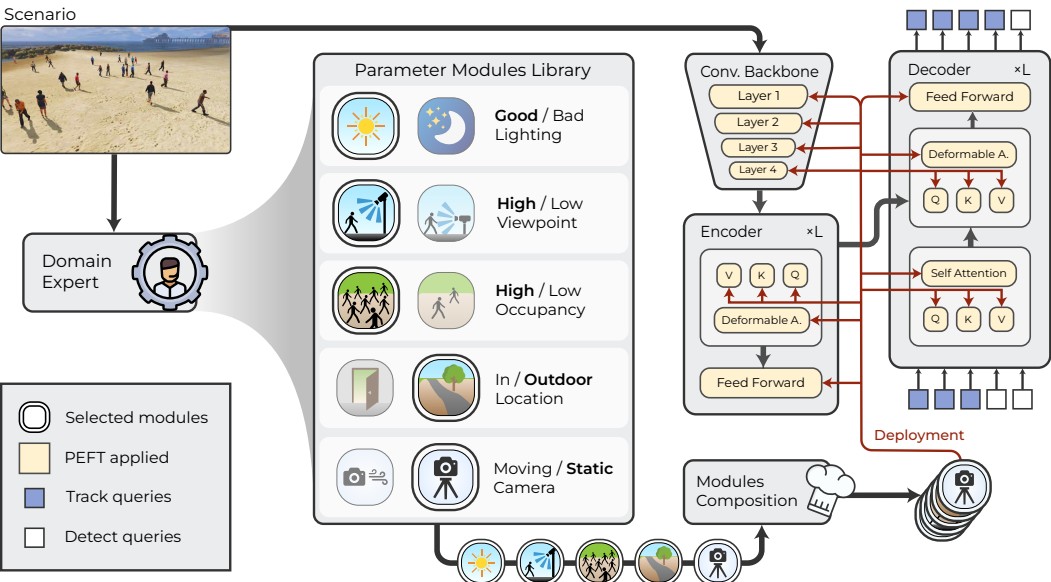

Figure 2: Overview of our modular architecture. A domain expert selects PEFT modules based on sequence attributes such as lighting and camera movement. These selected modules are then composed and applied to each model layer, adapting the backbone and encoder-decoder architecture.

**Query-based Multiple Object Tracking.** The underlying backbone of our transformer-based tracker follows the structure of [58]. In a nutshell, such a query-based model forces each query to recall the same instance across different frames. Specifically, we leverage an end-to-end trainable tracker built upon the Deformable DETR [6] framework conditioned by the image features extracted with a convolutional backbone (*i.e.*, ResNet [15]). Following [63], we further condition the DETR decoder with a set of detections from an external detector network and a shared learnable query.

At time $t = 0$, new proposals are generated from the objects detected in the scene. These proposals are then updated through self-attention and interact with image features via the deformable attention layer. The final prediction output is the summation of the initial anchors and the predicted offsets. For subsequent frames ($t > 0$), track queries generated from the previous frame are concatenated with learnable proposal queries of the current frame. Moreover, previous predictions are integrated with current proposals to establish new anchors for the incoming frame. We refer the reader to the original paper [63] for further details. It is noted that the flexibility of this architecture allows for the seamless integration of techniques based on modularity.

## 4 Method

We herein present PASTA, depicted in Fig. 2, a novel approach to Multiple Object Tracking that leverages PEFT modules to enable attribute-specific module specialization and reuse. This approach allows for the dynamic configuration of an end-to-end tracker by selecting the appropriate modules for each input scene, fully leveraging heterogeneous pre-training while avoiding negative transfer.

**Attribute-based modularity.** We devise a set of learnable modules to fine-tune each layer of our query-based tracker. Each module is related to an **attribute**: as shown in Fig. 3, we define $N = 5$ attributes, namely *lighting*, *viewpoint*, *occupancy*, *location*, and *camera motion*, and provide a tailored module for each discrete value these attributes take (see Sec. 5.3 for details). For instance, the *location* attribute has indoor and outdoor modules. At inference time, prior knowledge about the input scene is used to determine the appropriate value for each attribute, which in turn selects the corresponding modules from the "inventory", denoted as $M$.

Since the base model [63] relies on heterogeneous layers – namely, convolutional (*e.g.*, ResNet) and attention-based blocks (*e.g.*, Deformable DETR) – we employ two different strategies to fine-tune the modules. Specifically, after each convolutional layer of the ResNet backbone, we apply a strategy

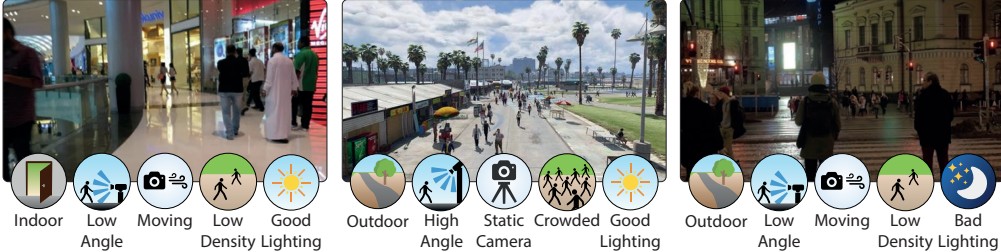

Figure 3: Examples of surveillance scenes and their corresponding attributes used by PASTA.

that learns channel-wise scale and shift parameters; for each layer of Deformable DETR, instead, we employ LoRA-based fine-tuning at each linear layer. In formal terms, considering each convolutional layer of the ResNet backbone, we deploy $|M|$ pairs $\{\gamma_m, \beta_m\}_{m=1}^{|M|}$ of learnable vectors $\gamma, \beta \in \mathbb{R}^C$, where $C$ is the number of the output channels. For each linear layer $l$ of the encoder-decoder structure underlying Deformable DETR, we devise $|M|$ pairs $\{A_m, B_m\}_{m=1}^{|M|}$ of learnable LoRA matrices.

During training, we start with the pre-trained weights and integrate all the modules while keeping the original parameters frozen. To prevent negative interference, we optimize each module *independently*, randomly sampling one attribute at a time and updating only the corresponding module at each training iteration. By the end of the training process, we obtain a set of specialized parameters (*experts*), which can be seamlessly merged during inference to improve overall tracking performance.

**Routing through Domain Expert.** During inference, two essential steps are required to exploit the learned modules: *routing* and *aggregation*. With multiple modules available from the inventory $M$, a routing strategy is required to determine the modules that should be active. To make this selection, we draw on what is known in the literature as *expert knowledge* [52, 35] (or "**Domain Expert**" in Fig. 2). In real-world applications such as video analytics, the expertise guiding the selection can come from a video surveillance operator or human analyst, who configures the appropriate modules to reflect domain- and scene-specific settings, such as camera perspective, lighting conditions, and other critical details. This approach allows users to optimize the tracking module for their unique contexts without extensive retraining. Additionally, the modular nature of the system enables easy integration of new modules to address emerging attributes or scenarios.

Relying on Domain Expert to select attributes is a grounded practice in real-world applications. For instance, the camera's mounting perspective and whether the scene is indoors or outdoors are typically known factors in fixed-camera scenarios. Additionally, automatic approaches can be envisioned to minimize human intervention further. For example, lighting conditions can be inferred by analyzing brightness levels, and a detector can count objects of interest in the scene, classifying crowd density.

**Modules composition.** In the final step, we aggregate the selected modules ("Modules Composition" in Fig. 2) and incorporate the result into the pre-trained tracker to create an expert model. Since these modules have been obtained by fine-tuning from $\boldsymbol{\theta}_0$, each module $\boldsymbol{\theta}^\star$ corresponds to a specific displacement $\tau^\star = \boldsymbol{\theta}^\star - \boldsymbol{\theta}_0$ in parameter space relative to the initial pre-training parameters $\boldsymbol{\theta}_0$. This displacement is known as the *task vector* [18]. The final composed model $f(\cdot; \boldsymbol{\theta}_c)$ is defined as:

$$f(\cdot; \boldsymbol{\theta}_c) \quad \text{where} \quad \boldsymbol{\theta}_c = \boldsymbol{\theta}_0 + \sum_{i=1}^{N} \lambda_i \tau_i, \quad \sum_i \lambda_i = 1 \text{ and } \tau_i \in M. \tag{1}$$

When $\lambda_i = \frac{1}{N}$, the formula simplifies to the average of the task vectors corresponding to each attribute. We employ this straightforward strategy for $\lambda_i$, giving equal weight to all attributes. However, considering the task vector $\tau_i$ associated with the $i$-th attribute, we employ a more sophisticated approach. If there are no domain shifts during inference (*i.e.*, both training and testing occur on the same dataset, such as MOTSynth), the task vector $\tau_i$ is simply set to the displacement $\tau^\star$ produced by the expert module selected by the Domain Expert. In contrast, when domain shifts are present (*e.g.*, training on MOTSynth and testing on MOT17), we adopt a soft strategy that considers *all* the modules in the inventory associated with the relevant attribute. In doing so, we follow the insights from [59], where the authors demonstrated that scenarios with shifting tasks benefit from richer representations than those derived from a single optimization episode.

Specifically, given the $i$-th attribute, let $R(i)$ be the set of its modules. We recall that each attribute admits multiple discrete values (*e.g.*, $R(\text{occupancy}) = \{\text{"low", "medium", "high"}\}$), and different

attributes may have different cardinalities (*e.g.*, $|R(\text{occupancy})| = 3$ and $|R(\text{lighting})| = 2$, as detailed in Sec. 5.3). Building on this, we employ soft routing to create the corresponding task vector, assigning the largest portion of the cake, *e.g.* $\rho = 0.80$, to the module selected by the Domain Expert. The remaining modules are weighted by $(1 - \rho)/(|R(i)| - 1)$, ensuring that the total sum equals 1. For example, considering those layers fine-tuned with the LoRA, the corresponding task vector is computed as:

$$\tau_i = \sum_{m \in R(i)} \bar{\lambda}_m B_m A_m, \quad \text{where} \quad \bar{\lambda}_m = \begin{cases} \rho & \text{if } m \text{ is selected,} \\ \frac{1 - \rho}{|R(i)| - 1} & \text{otherwise.} \end{cases} \tag{2}$$

Note that when $\rho = 1$, the soft strategy becomes hard, meaning that only the module selected by Domain Expert is utilized. By applying the formula above to all attributes, we obtain $N$ task vectors, which we aggregate following Eq. (1).

Similarly, we apply channel-wise scale and shift [23] operations to adapt each backbone layer. Formally, given the output $F$ of a convolutional layer, the $i$-th module applies a scale & shift operation to obtain the edited $\hat{F}_i$, such that $\hat{F}_i = \gamma_i \odot F + \beta_i$ with $\odot$ denoting the Hadamard product. At inference time, we combine the output of different scale & shift modules by noting that

$$\hat{F} = \sum_{i=1}^{N} \lambda_i(\gamma_i \odot F + \beta_i) = \sum_{i=1}^{N} \lambda_i(\gamma_i \odot F) + \lambda_i \beta_i = \left(\sum_{i=1}^{N} \lambda_i \gamma_i\right) \odot F + \sum_{i=1}^{N} \lambda_i \beta_i, \tag{3}$$

which means that parametrizing the scale & shift layer with a simple weighted average effectively results in averaging the outputs of the corresponding individual layers. The formula above applies to the in-domain setting but can be easily generalized to the soft routing scheme outlined by Eq. (2). Eventually, as discussed in [23], the scale & shift layer can be absorbed into the previous projection layer, thus ensuring that the inference process incurs no additional computational costs. The same re-parametrization trick can be employed to extract the task vector underlying scale & shift fine-tuning (refer to appendix A for additional notes).

## 5 Experiments

### 5.1 Datasets

**MOTSynth** [10] is a large synthetic dataset for pedestrian detection and tracking in urban scenarios, generated using a photorealistic video game. It comprises 764 full HD videos, each 1800 frames long, showcasing various attributes. In our experiments, following [29], we reduced the test sequences to 600 frames each and further split the training set to extract 48 validation sequences, shortened to 150 frames, for validation during training.

**PersonPath22** [44] is a large-scale pedestrian dataset consisting of 236 real-world videos featuring longer occlusions and more crowded scenes. It is divided into 138 training videos and 98 test videos.

**MOT17** [8] is a well-known benchmark, containing 7 sequences for training and 7 for testing, with different image resolutions, featuring crowded street scenarios with both static and moving cameras.

### 5.2 Experimental setting

We evaluate our proposed PASTA on both **in-domain** and **out-of-domain** scenarios. For the in-domain evaluation, we train and test PASTA on the MOTSynth synthetic dataset (Sec. 5.4) using expert modules in a domain-specific context. As a baseline, we train [63] on MOTSynth without using modules, referring to this model as MOTRv2-MS. For the out-of-domain evaluation, we conduct a synth-to-real zero-shot experiment on MOT17 and PersonPath22 (Sec. 5.5). Starting from training on MOTSynth, we test PASTA on these datasets without additional training, showcasing its ability to generalize under non-identically distributed domains. Finally, we present a series of ablation studies in Sec. 6 to take a closer look at the effectiveness of our method.

**Competing trackers and metrics.** In addition, we report the performance of other notable methods, including strong tracking-by-detection baselines such as ByteTrack [61] and OC-Sort [5]. We also include evaluations of query-based trackers, such as TrackFormer [31] and MOTRv2 [63] (see MOTRv2-MS). To compare their performance, we employ five metrics, ordered from detection to association, as recommended by [42]. These metrics are DetA [27], MOTA [2], HOTA [27], IDF1 [41], and AssA [27]. For the PersonPath22 dataset, we use their official metrics, MOTA and IDF1, supplemented by FP (false positives), FN (false negatives), and IDSW (identity switches).

Table 1: Evaluation on MOTSynth test set. $|\Theta|$ is the number of trainable parameters.

|  | $|\Theta|$ | HOTA↑ | IDF1↑ | MOTA↑ | DetA↑ | AssA↑ |
|---|---|---|---|---|---|---|
| SORT [3] | - | 46.0 | 55.7 | 50.9 | 49.9 | 42.8 |
| ByteTrack [61] | - | 45.7 | 56.4 | 61.8 | 50.1 | 41.9 |
| OCSort [5] | - | 46.9 | 56.8 | 59.1 | 48.7 | 45.6 |
| TrackFormer [31] | 44M | 41.3 | 49.9 | 47.7 | 44.4 | 40.6 |
| MOTRv2-MS | 42M | 52.4 | 56.6 | 61.9 | **56.4** | 49.0 |
| PASTA (*Ours*) | 15M | **53.0** | **57.6** | **62.0** | 56.2 | **50.4** |

## 5.3 Implementation details

We initialize our models using the pre-trained weights from DanceTrack [47], as provided by the authors of [63]. We employ YOLOX [14] as the auxiliary detector, exploiting weights from ByteTrack [61]. To provide a shared initialization for both PASTA and MOTRv2-MS training, we train a bootstrap model starting from the DanceTrack pre-train for 28k iterations on the MOTSynth training set. This bootstrap initialization uses half of the original training sequences from MOTSynth to align our model with the scenarios represented in the dataset. The learning rates are set to $5 \times 10^{-5}$ for the transformer and $1 \times 10^{-6}$ for the visual backbone.

In the second phase, we deploy the PEFT modules to fine-tune the bootstrap model. By excluding half the sequences during the bootstrap, we make sure that the modules can still learn valuable features. To ensure a fair comparison, we train each module for a similar number of iterations as MOTRv2-MS, with approximately 17k iterations. Regarding the encoder-decoder model, we apply our modularization strategy to every linear layer except those with output dimension less than 128. For the LoRA hyperparameters, we use $r = 16$, a weight decay of 0.1, and a learning rate of $3 \times 10^{-4}$. The scale & shift layers employ a learning rate of $1 \times 10^{-5}$ and a weight decay of $1 \times 10^{-4}$. The training is performed on a single RTX 4080 GPU with a batch size of 1 for both phases. Due to the small batch size, we accumulate gradients over four backward steps before performing an optimizer step. Each module is trained independently on the entire MOTSynth training set. With 12 modules, our model has approximately 15 million trainable parameters.

**Attributes.** We employ five key attributes to realize our modular architecture: lighting, camera viewpoint, people occupancy, location, and camera motion. For **lighting**, we specialize modules for *good* and *bad* lighting conditions. To do so, we threshold the brightness value V of the HSV representation at 70. The **viewpoint** attribute includes modules for *high*, *medium*, and *low* camera angles. We manually annotate this attribute as follows: *i)* scenes where the camera is parallel to the ground at or below pedestrian head level are labeled as "low-level"; *ii)* "high-level" viewpoints include vertical perspectives or scenes where the camera is positioned very high or far from people; and *iii)* "medium-level" includes all other camera angles. For **occupancy**, we design modules that reflect the crowd density within the scene: *low* (up to 10 people), *medium* (10 to 40 people), and *high* (more than 40 people), based on the count of detections with a confidence score above 0.2. The **location** attribute differentiates between *indoor* and *outdoor* settings. Lastly, the **motion** attribute comprises modules for both *moving* and *static* cameras, enabling the model to adapt to different camera movement scenarios. Further details on dataset statistics are provided in appendix C.

## 5.4 Performance in the in-domain setting

To assess the impact of the negative interference, we conduct several experiments on MOTSynth (see Tab. 1). Given the wide variety of scenarios in such a synthetic dataset, one can appreciate the advantages of using specialized modules. Indeed, integrating our modules resulted in an overall improvement w.r.t. its fine-tuning counterpart (MOTRv2-MS). Specifically, we observe an improvement over the association metrics (AssA, IDF1) and the HOTA and MOTA metrics. These enhancements suggest the benefits of our approach in reducing negative interference during training. By assigning each module a specific role tailored to particular scenario settings, we achieve improved training stability through a deterministic selection process guided by a domain expert.

Table 2: Zero-shot evaluation on MOT17. PASTA is evaluated in zero-shot by selecting the best attributes on the source dataset.

| | HOTA↑ | IDF1↑ | MOTA↑ | DetA↑ | AssA↑ |
|---|---|---|---|---|---|
| *fully-trained* | | | | | |
| SORT [3] | 64.3 | 73.1 | 70.9 | 63.3 | 66.1 |
| OC-SORT [5] | 66.4 | 77.8 | 74.5 | 64.1 | 69.1 |
| TrackFormer [31] | – | 74.4 | 71.3 | – | – |
| ByteTrack [61] | 67.9 | 79.3 | 76.6 | 66.6 | 69.7 |
| MOTRv2 [63] | 66.8 | 78.9 | 73.2 | 62.5 | 71.4 |
| *zero-shot* | | | | | |
| TrackFormer [31] | 51.0 | 63.9 | 58.7 | 51.8 | 61.2 |
| MOTRv2-MS | 62.6 | 73.0 | 67.6 | 60.3 | 65.5 |
| PASTA ($\rho = 1$) | 63.7 | 74.1 | 67.9 | 60.3 | 67.9 |
| PASTA ($\rho = 0.8$) | **64.0** | **74.9** | **68.1** | **60.4** | **68.3** |

Table 3: Evaluation on PersonPath22 test set. PASTA is evaluated in zero-shot by selecting the best attributes on the source dataset.

| | MOTA↑ | IDF1↑ | FP↓ | FN↓ | IDSW↓ |
|---|---|---|---|---|---|
| *fully-trained* | | | | | |
| CenterTrack [65] | 59.3 | 46.4 | 24 340 | 71 550 | 10 319 |
| SiamMOT [45] | 67.5 | 53.7 | 13 217 | 62 543 | 8942 |
| FairMOT [62] | 61.8 | 61.1 | 14 540 | 80 034 | 5095 |
| IDFree [46] | 68.6 | 63.1 | 9218 | 66 573 | 6148 |
| TrackFormer [31] | 69.7 | 57.1 | 23 138 | 47 303 | 8633 |
| ByteTrack [61] | 75.4 | 66.8 | 17 214 | 40 902 | 5931 |
| *zero-shot* | | | | | |
| TrackFormer [31] | 39.2 | 43.3 | 21 402 | 126 082 | 10023 |
| MOTRv2-MS | 48.3 | 53.1 | 28 483 | **98 007** | 7154 |
| PASTA ($\rho = 1$) | 49.7 | 53.7 | 18 211 | 105 611 | 6321 |
| PASTA ($\rho = 0.8$) | **50.0** | **53.8** | **18 038** | 105 454 | **6037** |

## 5.5 Performance in the out-of-domain setting

By designing distinct modules for various input conditions, we can effectively select the appropriate modules to handle distribution shifts, such as transitions to a new domain. We assess the benefits of this ability using synthetic data for training, and then evaluate on new, unseen datasets without any additional re-training (*zero-shot*). To do this, we start with our model trained on MOTSynth as described in Sec. 5.4 and evaluate it on MOT17 (Tab. 2) and PersonPath22 (Tab. 3). While these datasets share similarities in the attributes we employed, we emphasize that the source dataset is synthetic and the targets are real-world, resulting in a significant shift.

The results reported in Tab. 2 and 3 show an improvement over the baseline (*i.e.*, MOTRv2-MS), with +1.4 in HOTA and +1.9 in IDF1 in zero-shot MOT17, and +1.7 in MOTA and +0.7 in IDF1 in PersonPath22. Our approach demonstrates better generalization capabilities, helping close the gap with fully-trained methods while less computationally demanding. These results indicate that modularity enhances performance within the source dataset and improves domain generalization, leading to more reliable and versatile approach for tracking. Furthermore, other than reporting the results with the standard module selection (considering only the modules present in the scenes, $\rho = 1$), we also experiment with the weighted aggregation of all modules ($\rho = 0.8$) (detailed in Sec. 4). Interestingly, while the standard strategy shows improvements, the weighted aggregation strategy yields better performance. This suggests that richer representations, obtained by including multiple modules per attribute, are more effective for zero-shot scenarios than a single-module approach [59].

Table 4: Zero-shot evaluation of PASTA trained on MOT17 and tested on PersonPath22. PASTA is evaluated in zero-shot by selecting the best attributes on the source dataset.

| | MOTA↑ | IDF1↑ | FP↓ | FN↓ | IDSW↓ |
|---|---|---|---|---|---|
| *fully-trained* | | | | | |
| TrackFormer [31] | 69.7 | 57.1 | 23 138 | 47 303 | 8633 |
| ByteTrack [61] | 75.4 | 66.8 | 17 214 | 40 902 | 5931 |
| *zero-shot* | | | | | |
| MOTRv2-MS | 43.9 | 51.5 | 8304 | 119 391 | 5342 |
| PASTA | **46.1** | **54.6** | **7895** | **114 620** | **4702** |

Table 5: Ablation study on different module aggregation and selection strategies. (**Left**) MOTSynth validation, (**Right**) Zero-shot on MOT17 validation. The strategy we select is highlighted in yellow.

| MOTSynth (*val*) | HOTA↑ | IDF1↑ | MOTA↑ | MOT17 (*val*) | HOTA↑ | IDF1↑ | MOTA↑ |
|---|---|---|---|---|---|---|---|
| *aggregation* | | | | *aggregation* | | | |
| Sum (*only selected*) | 0.65 | 0.44 | -0.69 | Sum (only selected) | 0.58 | 0.41 | -0.03 |
| Weighted avg. ($\rho = 0.8$) | 59.9 | 66.8 | 59.6 | Weighted avg. ($\rho = 0.8$) | **64.0** | **74.9** | **68.1** |
| Mean avg. ($\rho = 1.0$) | **60.1** | **67.2** | **59.9** | Mean avg. ($\rho = 1.0$) | 63.7 | 74.1 | 67.9 |
| *selection* | | | | *selection* | | | |
| Opposite modules | 59.2 | 66.5 | 58.7 | Opposite modules | 62.9 | 73.9 | 67.1 |
| All modules | 59.8 | 67.0 | 59.4 | All modules | 63.1 | **74.1** | 67.7 |
| Domain Expert | **60.1** | **67.2** | **59.9** | Domain Expert | **63.7** | **74.1** | **67.9** |

**Evaluating zero-shot real-to-real transfer.** In Tab. 4, we present an additional experiment to evaluate the performance of PASTA in a zero-shot setting, this time using a realistic dataset as the source, rather than a synthetic one. For comparison, we train MOTRv2 on the MOT17 dataset and assess its performance on PersonPath22. Our approach showcases superior results compared to the fine-tuned MOTRv2, highlighting that leveraging modules enhances the model, with improved generalization capabilities in new and real-world domains.

## 6 Ablation studies

In Tab. 5, we evaluate the effect of various routing and aggregation strategies in both the in-domain setting (MOTSynth, left side of Tab. 5) and the zero-shot setting (MOT17, right side of Tab. 5). In the in-domain scenario, the results show that averaging the modules selected by the Domain Expert, specifically using Mean avg. ($\rho = 1.0$), is the most effective strategy. We also experimented with summation, as proposed by [60], but this method produced bad results, which we impute to the alteration of weight magnitudes when summing multiple modules. Another noteworthy approach is the *weighted avg.*, described in Sec. 4, which incorporates all modules, including those not selected.

While using only the *selected modules* is the optimal strategy in the in-domain scenario, for the zero-shot case (MOT17), incorporating knowledge from the non-selected modules — specifically, using Weighted avg. ($\rho = 0.8$) – enhances tracking performance. This pattern is also consistent when the domain shift involves evaluation on the PersonPath22 dataset (see appendix E).

**Module selection.** Should we select only the modules representing the current scenario, as determined by the Domain Expert approach, or would performance improve by incorporating all available modules? In Tab. 5, we investigate this matter by comparing these two approaches. To provide a more comprehensive perspective, we also evaluate a strategy that, in stark contrast to the Domain Expert, selects the *opposite modules* (*e.g.*, selecting the outdoor and poor lighting modules when presented with an indoor, well-lit scene). The lowest performance is observed when using opposite modules, indicating that using the proper modules provides valuable information about the current scene. Interestingly, the model still performs relatively well despite using opposite attributes, likely due to contributions from other modules whose general knowledge of the domain sustains overall performance. This suggests that modules can assist one another in solving tasks. Moreover, reduced

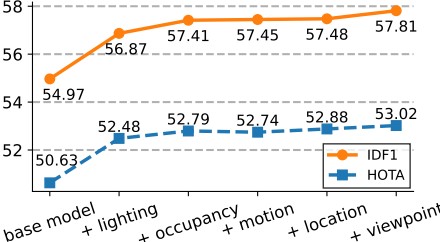

Figure 4: IDF1 and MOTA when adding new attributes on MOTSynth.

Table 6: Opposite modules selection.

| | HOTA↑ | IDF1↑ | MOTA↑ |
|---|---|---|---|
| No modules | 58.2 | 64.9 | 55.6 |
| Opposite: only lighting | 58.9 | 66.2 | 57.8 |
| Opposite: only viewpoint | 58.8 | 65.7 | 57.0 |
| Opposite: only occupancy | 58.6 | 66.4 | 59.2 |
| Opposite: only location | 58.9 | 66.0 | 58.4 |
| Opposite: only camera | 58.5 | 65.6 | 58.4 |
| Average opposite | 58.7 | 66.0 | 58.2 |
| Correct modules | **60.1** | **67.2** | **59.9** |

Table 7: Performance comparison of our approach without applying fine-tuning and to specific parts of the architecture (*i.e.*, decoder, encoder, visual backbone).

| Fine-tuning applies on | HOTA↑ | IDF1↑ | MOTA↑ | DetA↑ | AssA↑ |
|---|---|---|---|---|---|
| none | 52.4 | 56.6 | 61.9 | **56.4** | 49.0 |
| all except the decoder | 51.5 | 56.0 | 58.9 | 53.6 | 49.8 |
| all except the encoder | 52.4 | 56.9 | 61.2 | 55.7 | 49.7 |
| all except the backbone | 52.5 | 57.0 | 61.5 | 55.6 | 49.9 |
| PASTA (all) | **53.0** | **57.6** | **62.0** | 56.2 | **50.4** |

negative interference – achieved by training each module separately – prevents the modules from relying on each other and allows them to make unique contributions independently.

Furthermore, in Fig. 4, we illustrate how the incremental addition of specialized modules improves IDF1 and HOTA metrics, showcasing that greater specialization of the modules gradually enhances overall performance. For a more detailed analysis, in Tab. 6, we select the opposite modules instead of the correct one for each attribute. Although the metrics are further reduced, the model performs well due to its robust pre-training, as indicated by the *no modules* baseline shown in the table.

**Block-wise analysis.** In our approach, attribute-related modules are applied to edit the entire network. However, users may opt to edit selectively specific parts of the architecture, thereby identifying which components are most critical. In Tab. 7, we conduct an ablation study by excluding our modules from being applied to varying components of the architecture. The results indicate that not applying task vectors to the decoder significantly degrades detection and association metrics. We believe that this degradation can be explained by considering the crucial role of the decoder. The decoder must indeed gather information from detection, tracking, and proposal queries while simultaneously integrating visual information from the encoder. Consequently, not adapting the decoder prevents the architecture from effectively leveraging queries and visual cues. The encoder also contributes substantially, though to a lesser extent than the decoder, as it primarily refines and contextualizes visual features from the backbone. Finally, the backbone shows the smallest contribution.

## 7 Conclusions

In this work, we introduce PASTA, a novel framework that enhances domain generalization in tracking-by-query methods for Multiple Object Tracking. Our approach features a modular structure with dedicated modules tailored to different attributes of real-world scenes. These modules utilize Parameter-Efficient Fine-Tuning techniques, enabling the integration of scene-specific parameters while minimizing computational load. Comprehensive experiments demonstrate that domain-specialized modules significantly bolster robustness, allowing effective adaptation across domains without extensive retraining. PASTA further enables camera operators to configure the optimal module for each unique scenario, ensuring precise adaptation to diverse real-world settings.

## Acknowledgements

The research activities conducted by Angelo Porrello were funded by the Italian Ministry for University and Research under the PNRR project ECOSISTER ECS 00000033 CUP E93C22001100001. Additionally, the research carried out by Rita Cucchiara was supported by the EU Horizon project "ELIAS - European Lighthouse of AI for Sustainability" (No. 101120237).

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

# Appendix

## A  Extracting task vectors from scale & shift layers

In our convolutional backbone, we apply a channel-wise scale and shift [23] operation to adapt each backbone layer. We provide details on computing the task vector as an increment relative to the base parameters. Following [23], this is achieved by re-parameterizing the scale and shift as:

$$\hat{F} = \left(\sum_{m \in R(i)} \bar{\lambda}_m \gamma_m\right) \odot (W_0 * h + b_0) + \left(\sum_{m \in R(i)} \bar{\lambda}_m \beta_m\right), \tag{A}$$

$$= \underbrace{\left(\sum_{m \in R(i)} \bar{\lambda}_m \gamma_m \odot W_0\right)}_{W^\star} * h + \underbrace{\sum_{m \in R(i)} \bar{\lambda}_m \left(\gamma_m \odot b_0 + \beta_m\right)}_{b^\star}. \tag{B}$$

where $W_0 \in \mathbb{R}^{C_{\text{out}} \times C_{\text{in}} \times H \times W}$ and $b_0 \in \mathbb{R}^{C_{\text{out}}}$ represent the original convolution weights and bias, $C_{\text{out}}$ is the number of convolution output channels, $C_{\text{in}}$ is the number of convolution input channels, $H$ and $W$ are the height and width of the kernel, $h$ is the output of the preceding layer, and $*$ denotes the convolution operation. Notice that, to simplify the notation, we assume to reshape $\gamma_m \in \mathbb{R}^{C_{\text{out}} \times 1 \times 1 \times 1}$ and implicitly broadcast in accordance with the dimensions of $W_0$ before applying the Hadamard product $\odot$.

The task vectors $\tau_\gamma = W^\star - W_0$ and $\tau_\beta = b^\star - b_0$ for the scale and shift parameters are defined:

$$\tau_\gamma = \left(\sum_{m \in R(i)} \bar{\lambda}_m \gamma_m \odot W_0\right) - W_0, \tag{C}$$

$$\tau_\beta = \sum_{m \in R(i)} \bar{\lambda}_m \left(\gamma_m \odot b_0 + \beta_m\right) - b_0. \tag{D}$$

By representing our attributes as task vectors and leveraging pre-computed weights, we ensure that the inference process incurs no additional computational costs.

## B  Dataset licenses

- **MOTSynth** is released under the MIT License.
- **MOT17** is released under the CC BY-NC-SA 3.0 License.
- **PersonPath22** is released under the CC BY-NC 4.0 License.

## C  Dataset statistics

Tab. A presents statistics on the employed datasets, detailing attributes at both per-sequence and per-frame levels. We manually annotated these attributes, developing a custom annotation tool that displays the first frame of each sequence and allows for efficient annotation using keybindings. This process required minimal effort, involving one annotator for approximately three hours on MOTSynth and two hours on PersonPath22. As shown in Tab. A, the statistics indicate an imbalance in certain attributes; to address this, we implemented a custom training sampler to ensure that each module receives an equal number of backward iterations.

## D  Forgetting on source dataset

Recently, there has been growing interest in Continual Learning for large pre-trained models, particularly in incrementally fine-tuning these models using parameter-efficient methods [11, 64, 57]. A key challenge in this process is avoiding the issue of catastrophic forgetting [30], where a model loses knowledge from earlier training as new tasks are introduced. To this end, we evaluated the extent of forgetting in the model when using task-specific modules versus training the entire model. Namely, we start from the PASTA and MOTRv2-MS trained on MOTSynth as in Tab. 1. Then, we further fine-tune such models on MOT17 and evaluate again on MOTSynth to measure the source-domain performance after the adaptation. As shown in Tab. B, the modular approach trained on MOT17 is less prone to forget its pre-training on MOTSynth, achieving superior results compared to full fine-tuning when tested again on MOTSytnh test split. Specifically, our modular approach PASTA

Table A: Per-sequence and per-frame attributes statistics in PersonPath22, MOT17, and MOTSynth.

| | PersonPath22 | MOT17 | MOTSynth |
|---|---|---|---|
| *Per-sequence attributes* | | | |
| **Total Sequences** | **236** | **12** | **764** |
| Indoor | 61 | 2 | 57 |
| Outdoor | 175 | 12 | 707 |
| Camera Low | 162 | 10 | 479 |
| Camera Mid | 68 | 4 | 244 |
| Camera High | 6 | 0 | 41 |
| Moving | 60 | 8 | 220 |
| Static | 176 | 6 | 544 |
| Bad Light | 23 | 0 | 189 |
| Good Light | 213 | 14 | 575 |
| *Per-frame attributes* | | | |
| **Total Frames** | **203653** | **5316** | **1375200** |
| Occupancy Low | 44% | 24% | 29% |
| Occupancy Mid | 53% | 54% | 68% |
| Occupancy High | 3% | 22% | 3% |

Table B: Source-domain (MOTSynth) results before and after fine-tuning on target-domain (MOT17). We report the difference in performance in brackets.

| MOTSynth | HOTA↑ | IDF1↑ | MOTA↑ | DetA↑ | AssA↑ |
|---|---|---|---|---|---|
| *Trained on MOTSynth (Tab. 1)* | | | | | |
| MOTRv2-MS | 52.4 | 56.6 | 61.9 | **56.4** | 49.0 |
| PASTA | **53.0** | **57.6** | **62.0** | 56.2 | **50.4** |
| *Subsequently trained on MOT17* | | | | | |
| MOTRv2-MS | 48.1 **(-4.3)** | 56.3 **(-0.3)** | 60.8 **(-1.1)** | 50.7 **(-5.7)** | 46.2 **(-2.8)** |
| PASTA | **49.8 (-3.2)** | **57.4 (-0.2)** | **61.8 (-0.2)** | **52.3 (-3.9)** | **48.0 (-2.4)** |

outperforms the standard one across all metrics, demonstrating the effectiveness of the modular training in mitigating catastrophic forgetting. Indeed, the LoRA [16] modules act as a regularizer that mitigates forgetting of the source-domain [4].

# E  Additional ablation studies

To further support our claim on leveraging all modules in a zero-shot scenario, we conduct an additional ablation study on the test split of PersonPath22. As shown in Tab. C, this test confirms that retaining knowledge from modules not directly related to the specific scenario is beneficial when dealing with domain shifts. Specifically, our selection strategy outperforms the unweighted average by empirically assigning a weight $\rho = 0.8$ to the selected modules and $\rho = 0.2$ to the others.

**Comparison with tracking-by-detection.** To comprehensively evaluate our method, we herein test ByteTrack and other tracking-by-detection methods in a zero-shot setting from MOT17 to PersonPath22. We report the results on PersonPath22 in Tab. D. Results indicate that PASTA leads to remarkable improvements compared to the other query-based end-to-end approach (*i.e.*, MOTRv2 [63]), even though they are both outperformed by the tracking-by-detection methods (such as ByteTrack [61]). To be more comprehensive, PASTA remains competitive in terms of association performance (IDF1), but it yields weaker detection capabilities. Such a trend does not surprise us and is in line with what occurs in the more standard evaluation, where fine-tuning on the target dataset is allowed. Indeed, tracking-by-detection approaches are generally more robust than those based on

Table C: Ablation study on different module aggregation strategies on PersonPath22 test set in zero-shot.

| PersonPath22 | MOTA↑ | IDF1↑ | FP↓ | FN↓ | IDSW↓ |
|---|---|---|---|---|---|
| Sum (only selected) | 0.91 | 0.64 | – | – | – |
| Avg. (only selected) | 49.6 | 53.6 | 18 211 | 105 611 | 6321 |
| **Weighted avg. (all)** | **50.0** | **53.8** | **17 786** | **105 454** | **6037** |

Table D: Comparison with tracking-by-detection approaches in a zero-shot setting from MOT17 to PersonPath22.

| | Setting | IDF1↑ | MOTA↑ | FP↓ | FN↓ | IDSW↓ |
|---|---|---|---|---|---|---|
| ByteTrack | fine-tuned on PP22 | 66.8 | 75.4 | 17 214 | 40 902 | 5931 |
| ByteTrack | zero-shot | 56.2 | 55.9 | 3307 | 106 892 | 3962 |
| OC-SORT | zero-shot | 55.6 | 59.9 | 3254 | 94 786 | 5786 |
| SORT | zero-shot | 48.5 | 57.4 | 40 173 | 56 003 | 14 060 |
| MOTRv2-MS | zero-shot | 43.9 | 51.5 | 8304 | 119 391 | 5342 |
| PASTA | zero-shot | **46.1** | **54.6** | **7895** | **114 620** | **4702** |

end-to-end learning, so much so that it is an established practice to present the results in separate parts of a table [12, 58, 63], to deliver an apple-to-apple comparison.

In a zero-shot setting, we conclude that existing tracking-by-detection trackers are more robust to domain shifts. In these approaches, the only component potentially subject to shifts is the detector (*e.g.*, YOLOX [14]). Instead, the motion model (*e.g.*, Kalman Filter [19]) and the association strategy [3, 61] are almost parameter-free procedures that are less affected by domain shifts for construction, as their design reflects strong inductive biases about human motion. For such a reason, it is our belief that the problem of domain shift in Multiple Object Tracking (MOT) should be primarily addressed in parametric approaches such as deep neural networks. For this reason, our research question focuses on query-based trackers (e.g., MOTRv2) that learn entirely from data. Our final goal is to enhance these trackers, as their end-to-end nature results can lead to challenges during domain shifts.

**ByteTrack thresholds.** In Tab. 1, we evaluated ByteTrack on MOTSynth using the default input-parameters provided in the public ByteTrack repository, specifically a minimum confidence score (`min_score`) of $0.1$ and a track threshold (`track_thresh`) of $0.6$. In Tab. E, we present the results for different values of these thresholds. The results are close, with a slight improvement when reducing the `track_thresh` to $0.3$ or $0.4$, while `min_score` of $0.1$ remains optimal.

# F   On computational costs

**Memory efficiency.** We compare the GPU memory of full fine-tuning versus our approach on the MOTSynth dataset. Our method reduces training GPU memory requirements from 13GB to 8.25GB (for a batch size of 1), a reduction of over 35%. This significant decrease is due to the lower number of parameters updated by the optimizer: 42M parameters for standard fine-tuning versus 15M for our PEFT technique, as reported in Tab. 1.

**Inference speed.** Additionally, our approach does not add any overhead during inference, aside from weight merging, which is negligible for stationary attributes, compared to MOTRv2, which maintains a speed of 6.9 FPS on a 2080Ti GPU.

**Storage efficiency.** Using PEFT techniques significantly reduces storage needs. Without these techniques, each attribute would require a fully fine-tuned model, which poses several challenges, especially in memory constraints [16, 25]. Firstly, storing a separate model for each attribute is highly storage-intensive. For instance, a PASTA module is approximately 5MB, whereas the full model exceeds 350MB. With 12 attributes, the total storage requirement for PASTA would be 410MB

Table E: ByteTrack thresholds sensitivity analysis on MOTSynth.

| | Score | Threshold | HOTA↑ | IDF1↑ | MOTA↑ | DetA↑ | AssA↑ |
|---|---|---|---|---|---|---|---|
| | 0.1 | 0.2 | 45.9 | 56.4 | 61.9 | 51.1 | 41.6 |
| | 0.1 | 0.3 | 46.0 | 56.5 | 62.1 | 51.1 | 41.7 |
| | 0.1 | 0.4 | 45.9 | 56.6 | **62.2** | 50.9 | 41.8 |
| ByteTrack | 0.05 | 0.6 | 43.0 | 54.4 | 54.5 | 45.9 | 40.9 |
| | 0.1 | 0.6 | 45.7 | 56.4 | 61.8 | 50.1 | 41.9 |
| | 0.2 | 0.6 | 45.6 | 56.3 | 61.5 | 49.8 | 41.9 |
| | 0.1 | 0.7 | 45.0 | 55.8 | 60.3 | 48.7 | 41.8 |
| PASTA (*Ours*) | - | - | **53.0** | **57.6** | 62.0 | **56.2** | **50.4** |

(350MB + 12 x 5MB). In contrast, storing 12 fully fine-tuned models would require around 4.2GB (12 x 350MB), representing a tenfold increase in storage needs. Additionally, adapting an entire model to each specific condition is more time-consuming than using LoRA, as it involves optimizing a more significant number of parameters. This adaptation process must be repeated for each attribute, making it both impractical and costly. Moreover, fully fine-tuning a transformer-based architecture demands more data than a parameter-efficient approach.

# G   Limitations

One limitation of our approach is the reliance on an expert router, which requires manual data annotation or intervention by an external domain expert. This process can be resource-intensive and may not scale well for larger datasets or diverse scenarios. Future work may explore the development of automatic routing techniques, which could significantly improve scalability, performance, and ease of deployment in real-world applications by reducing the dependency on manual annotations.

# H   Societal impacts

**Positive Impacts.**   Enhanced security and surveillance is one of the key benefits of this work. Improved accuracy and robustness in tracking can lead to better crime prevention, more efficient law enforcement, and increased public safety. Additionally, operational efficiency is another positive impact, where various sectors, including transportation, retail, and urban planning, can benefit from optimized operations and resource allocation. Moreover, customization and adaptability are enhanced by tailoring modules for specific scenarios, increasing versatility in applications ranging from healthcare to sports analytics.

**Negative Impacts.**   However, there are also potential negative impacts to consider. Privacy concerns arise from increased tracking capabilities, which may lead to unauthorized surveillance and privacy infringement. Bias and fairness are also issues, as biased training data can perpetuate existing biases, leading to unfair treatment of certain groups.

While the modular approach presents significant advancements, it is crucial to address these societal impacts through careful design and transparent policies.

