# OpenReview forum: "Is Multiple Object Tracking a Matter of Specialization?"
_NeurIPS.cc/2024/Conference — NeurIPS 2024 poster_

### Official Review · Reviewer_Vx5C · 2024-07-10

**Soundness:** 3
**Presentation:** 3
**Contribution:** 2
**Rating:** 5
**Confidence:** 5

**Summary:**

The paper introduces PASTA, a framework designed to address the challenges of training end-to-end transformer-based trackers in heterogeneous scenarios, specifically negative interference and poor domain generalization. PASTA leverages Parameter-Efficient Fine-Tuning (PEFT) and Modular Deep Learning (MDL) to define and train specialized modules based on key scenario attributes like camera viewpoint and lighting conditions. These modules are then combined using task arithmetic to enhance generalization to new domains. The framework's effectiveness is demonstrated through extensive experiments on MOTSynth and zero-shot evaluations on MOT17 and PersonPath22, where it outperforms traditional monolithic trackers. The primary contributions include the introduction of PASTA, the use of PEFT and MDL for better domain adaptation, and the validation of its superior performance in diverse tracking scenarios.

**Strengths:**

The paper introduces an innovative framework that combines Parameter-Efficient Fine-Tuning (PEFT) and Modular Deep Learning (MDL) to tackle challenges faced by end-to-end transformer-based trackers in heterogeneous scenarios, setting it apart from traditional approaches. The research quality is robust, with extensive experiments on MOTSynth, MOT17, and PersonPath22 validating the framework's effectiveness. The paper is well-structured and clearly explains key concepts and methodologies, making it accessible and easily understood.

**Weaknesses:**

The paper has several weaknesses that should be addressed.
1. The authors do not compare their framework to ByteTrack and other trackers in zero-shot settings, limiting their evaluation's comprehensiveness.
2. For a fair comparison in Table 1, the authors should provide detailed information on the different detection thresholds used, as varying thresholds can significantly affect ByteTrack's performance.
3. The authors need to discuss their choice of attributes in more detail, explaining why specific attributes were selected and their relevance to the framework.
4. It remains unclear if the performance could be improved without using LoRA, as the authors do not explore or report the impact of omitting LoRA in their experiments.

**Questions:**

I have listed my concerns and questions above.

**Limitations:**

The authors have claimed the limitations.

---

> ### Author Rebuttal · Authors · 2024-08-05
>
> **W1 - Comparison with other methods in zero-shot**
>
> We herein test ByteTrack and other tracking-by-detection methods in a zero-shot setting from MOT17 to PersonPath22. The results on PersonPath22 are as follows:
>
> |Tracker|Setting|IDF1|MOTA|FP|FN|IDsw|
> |-|-|-|-|-|-|-|
> |ByteTrack|fine tuned on PersonPath22|66.8|75.4|17214|40902|5931|
> |ByteTrack|zero shot results|56.2|55.9|3307|106892|3962|
> |OC-SORT|zero shot results|55.6|59.9|3254|94786|5786|
> |SORT|zero shot results|48.5|57.4|40173|56003|14060|
> |MOTRv2|zero shot results|51.5|48.6|8304|119391|5342|
> |PASTA|zero shot results|54.6|50.8|7895|114620|4702|
>
> Results indicate that PASTA leads to remarkable improvements compared to the other query-based end-to-end approach (i.e., MOTRv2), even though they are both outperformed by the tracking-by-detection methods (such as ByteTrack). To be more comprehensive, PASTA remains competitive in terms of association performance (IDF1), but it yields weaker detection capabilities. Such a trend does not surprise us and is in line with what occurs in the more standard evaluation, where fine-tuning on the target dataset is allowed. Indeed, tracking-by-detection approaches are generally more robust than those based on end-to-end learning, so much so that it is an established practice of researchers to present the results in separate parts of a table [10, 43, 48], to deliver an apple-to-apple comparison.
>
> Concerning the zero-shot setting in our work, we conclude that existing tracking-by-detection trackers are more robust to domain shifts. In these approaches, the only part potentially subject to shifts is the detector (e.g., YOLOX). Instead, the motion model (e.g., Kalman Filter) and the association strategy [3, 46] are almost parameter-free procedures that are less affected by domain shifts for construction, as their design reflects strong inductive biases about human motion. For such a reason, it is our belief that the problem of domain shift in Multiple Object Tracking (MOT) should be primarily addressed in parametric approaches such as deep neural networks. For this reason, our research question focuses on query-based trackers (e.g., MOTRv2) that learn entirely from data. Our final goal is to enhance these trackers, as their end-to-end nature results can lead to challenges during domain shifts. We will include such a discussion in the final manuscript.
>
> **W2 - Detection Thresholds used in Tab 1**
>
> We evaluated ByteTrack on MOTSynth using the default values provided in the public ByteTrack repository, specifically a minimum confidence score (min_score) of 0.1 and a track threshold (track_thresh) of 0.6. Below, we present the results for different values of these thresholds:
>
> ||HOTA|IDF1|MOTA|DetA|AssA|
> |-|-|-|-|-|-|
> |_ByteTrack (min_score: 0.1, track_thresh: 0.6)_|45.7|56.4|61.8|50.1|41.9|
> |ByteTrack (min_score: 0.05, track_thresh: 0.6)|43.0|54.4|54.5|45.9|40.9|
> |ByteTrack (min_score: 0.2, track_thresh: 0.6)|45.6|56.3|61.5|49.8|41.9|
> |ByteTrack (min_score: 0.1, track_thresh: 0.2)|45.9|56.4|61.9|51.1|41.6|
> |ByteTrack (min_score: 0.1, track_thresh: 0.3)|46.0|56.5|62.1|51.1|41.7|
> |ByteTrack (min_score: 0.1, track_thresh: 0.4)|45.9|56.6|62.2|50.9|41.8|
> |ByteTrack (min_score: 0.1, track_thresh: 0.7)|45.0|55.8|60.3|48.7|41.8|
> |PASTA|53.0|57.6|62.0|56.2|50.4|
>
> The results are close, with a slight improvement when reducing the track_thresh to 0.3 or 0.4, while min_score of 0.1 remains optimal. We will report these updated results in the final version, along with more details about the parameters used.
>
> **W3 - Choice of attributes**
>
> We selected the attributes reported because we believe they are the most generic and applicable to various scenarios. However, users are not limited to using these specific attributes; PASTA is fully customizable to suit different needs. For instance, if users know their system will always be employed outdoors, they could omit the indoor/outdoor attributes and add a good/poor weather attribute instead. This flexibility allows PASTA to be tailored to specific requirements, enhancing its adaptability and effectiveness across various applications. This adaptability ensures that users can optimize the trakcer for their unique contexts without needing extensive retraining or adjustments.
>
> Furthermore, PASTA's modular nature means that users can easily integrate additional modules to handle these changes as new attributes become relevant or new scenarios emerge. This future-proofs the system and allows it to evolve alongside technological advancements and shifting user needs. For example, attributes such as traffic density or driving side could be added in an automotive setting to improve performance. The ability to customize and extend PASTA ensures it remains a robust and versatile tool for a wide range of tracking applications.
>
> **W4 - Omitting LoRA**
>
> Omitting LoRA (or other Parameter-Efficient Fine-Tuning techniques) in PASTA would significantly impact its efficiency in scenario-specific adaptation. Without these techniques, each attribute would require a fully fine-tuned model, which poses several challenges, especially memory constraints [12,19].
> Firstly, storing a separate model for each attribute is highly storage-intensive. For instance, a PASTA module is approximately 5MB, whereas the full model exceeds 350MB. With 12 attributes, the total storage requirement for PASTA would be 410MB (350MB + 12 x 5MB). In contrast, storing 12 fully fine-tuned models would require around 4.2GB (12 x 350MB), representing a tenfold increase in storage needs.
> Additionally, adapting an entire model to each specific condition is more time-consuming than using LoRA, as it involves optimizing a more significant number of parameters. This adaptation process must be repeated for each attribute, making it both impractical and costly. Moreover, fully fine-tuning a transformer-based architecture demands more data than a parameter-efficient approach.

---

> ### Comment · Reviewer_Vx5C · 2024-08-11
>
> Thanks for your response. The authors address my main concerns.

---

### Official Review · Reviewer_4fn5 · 2024-07-11

**Soundness:** 3
**Presentation:** 3
**Contribution:** 3
**Rating:** 5
**Confidence:** 5

**Summary:**

This paper proposes a new framework called PASTA (Parameter-Efficient Scenario-specific Tracking Architecture), which aims to improve the generalization ability of multi-object tracking (MOT) in diverse scenarios. The main contributions of this paper include: 1) proposing the PASTA framework to achieve efficient query tracker fine-tuning through PEFT technology; 2) improving domain transfer and preventing negative interference by introducing expert modules; 3) validating the method's effectiveness in zero-shot tracking scenarios through a comprehensive evaluation.

**Strengths:**

1，The paper thoroughly verifies the effectiveness of the PASTA framework through extensive experiments on multiple datasets (MOTSynth, MOT17, PersonPath22). These experimental results clearly demonstrate the significant advantages of PASTA in reducing negative interference and improving domain generalization ability, particularly its excellent performance in zero-shot settings. On the MOTSynth test set, PASTA shows improvements in multiple key metrics (such as HOTA, IDF1, MOTA, DetA, AssA) compared to MOTRv2-MS, proving the advantage of its modular design in handling complex scenarios.
2，The paper is structured tightly and logically, with each part (introduction, methodology, experiments, and conclusion) clearly laid out, making it easy for readers to follow and understand. The methodology section, in particular, provides detailed descriptions of the design principles and implementation steps of the PASTA framework, allowing readers to clearly grasp its working principles and innovations.
3，The paper uses professional and accurate academic language, clearly expressing the research objectives, methods, and results. The concise and clear description of technical details and experimental results enhances the paper's persuasiveness and credibility.
4，The figures and tables in the paper are exquisitely designed, visually presenting experimental results and method structures. For example, Figure 1 shows the modular architecture of PASTA, allowing readers to intuitively understand the combination and application of different modules. The experimental result tables are also clear, facilitating the comparison of different methods' performances.
5，The PASTA framework combines Parameter-Efficient Fine-Tuning (PEFT) and Modular Deep Learning (MDL) technologies, proposing a new multi-object tracking solution. By defining key scenario attributes and training specialized PEFT modules for each attribute, PASTA performs excellently in handling heterogeneous scenarios, demonstrating significant innovation and research value. The paper introduces the concept of combining modules using task arithmetic, significantly reducing negative interference and enhancing domain generalization ability, showcasing the potential of modular design in deep learning.

**Weaknesses:**

1，The PASTA framework relies on manual selection by domain experts to determine the appropriate modules for the current scenario. Although effective, this method has limitations in practical applications. In some scenarios, it may not always be possible to obtain support from domain experts, or the judgments of domain experts may be influenced by subjective factors, affecting the overall performance of the model. Additionally, this manual selection method may lead to consistency and repeatability issues and is challenging to automate in large-scale applications.
2，Although the PASTA framework has been extensively tested on the MOTSynth, MOT17, and PersonPath22 datasets, these experiments mainly focus on pedestrian tracking and surveillance scenarios. There is a lack of validation in other important application fields (such as traffic monitoring, smart retail, autonomous driving, etc.), limiting the generality and promotion of the results. Further experiments can help verify the application effects of PASTA in broader scenarios, enhancing its generality and practical value.
3，The paper mentions that PASTA reduces computational costs through parameter-efficient fine-tuning, but it lacks detailed analysis of specific computational resources and time costs. For example, in practical applications, how much computational resources (such as GPU/CPU time, memory requirements, etc.) are specifically needed, and how much time is saved compared to traditional methods. Detailed analysis of resources and time costs can help evaluate the practical feasibility of PASTA in different application scenarios.
4，Although the paper mentions the advantages of modular design, it lacks an in-depth study of the potential synergy effects between different module combinations. There may be complementary or conflicting effects between different modules, which have significant impacts on the final performance. Conducting relevant experiments and analysis can help understand which module combinations are most effective, thereby optimizing the design and application strategies of PASTA, further enhancing model performance.
5，The paper does not provide a detailed discussion on the robustness of model parameters in the PASTA framework. There is a lack of experiments testing the model's performance under different noise levels, data quality, and data volumes. These tests can verify the model's stability and robustness in practical applications, further improving the credibility and practicality of PASTA. These experiments can demonstrate the adaptability and stability of PASTA in various real-world environments, ensuring it maintains high performance under various conditions.

**Questions:**

1，The modular design proposed in the paper effectively avoids negative interference and improves domain generalization ability. However, the selection and combination of modules can be further optimized using more intelligent methods, such as reinforcement learning or other automated techniques, instead of relying on manual selection by domain experts. Intelligent optimization of module selection can further enhance the model's adaptability in complex scenarios while reducing dependence on domain experts, thereby increasing the practical application value of the system.
2，The paper verifies the effectiveness of PASTA on multiple datasets, but it should consider validation in more real-world application scenarios, such as traffic monitoring, smart retail, autonomous driving, etc. These fields have unique tracking needs and challenges. By verifying PASTA in these scenarios, the wide applicability and practical value of PASTA can be further demonstrated, enhancing the paper's persuasiveness and showcasing PASTA's potential application effects in different fields.

3， When combining modules, the synergy between modules should be considered, and related experiments should be added to explore the performance impact of different module combinations. Specifically, experiments can be designed to evaluate which module combinations can produce the best effects or which combinations may lead to performance degradation. In-depth research on module synergy can help understand and optimize the PASTA module design, improve overall model performance, and provide valuable references for future module development.
4，Discussion of Computational Resources and Time Costs： Although PASTA reduces computational costs through parameter-efficient fine-tuning, the specific computational resources and time costs in practical applications have not been discussed in detail. Particularly in comparison with traditional methods, clearly listing the specific computational resources (such as GPU/CPU time, memory requirements, etc.) and time costs required in the training and inference stages can help readers better understand the feasibility and advantages of PASTA in practical applications. This analysis is crucial for evaluating the practical benefits of PASTA in large-scale real-world applications.
5，The discussion of future work and application prospects of the PASTA framework is relatively brief and does not fully demonstrate its long-term value and expansion potential.

**Limitations:**

1，Complexity of Modular Systems
The modular design of the PASTA framework increases the complexity of the system, especially in terms of training, managing, and combining modules. Each module requires separate training, optimization, and validation, increasing development and maintenance costs. In practical deployment, modular systems require more complex architectural support, involving dynamic selection and combination of modules, which may increase implementation difficulty and system maintenance complexity.
2，Generalization Limitations
Although PASTA performs well on specific datasets and scenarios, its design and optimization target specific scenario attributes (such as camera angles, lighting conditions, etc.). PASTA's generalization and adaptability may be limited when dealing with new scenarios that do not belong to these predefined attributes. The framework's performance in completely different application environments or new domains has not been fully validated, potentially affecting its promotion and widespread application.
3，Dataset Dependency
The experiments in the paper are mainly based on the MOTSynth, MOT17, and PersonPath22 datasets, which may have certain biases or characteristics. The excellent performance of the PASTA framework on these datasets may not be directly generalizable to other datasets or more diverse data sources. The lack of experimental validation on more diverse datasets limits the comprehensive evaluation of PASTA's generalization ability in various scenarios.
4，Dependence on Pre-trained Models
The overall performance of the PASTA framework largely depends on pre-trained backbone networks and detectors. If pre-trained models perform poorly in certain tasks or scenarios, the performance of the PASTA framework will also be significantly affected. This means that the application and effectiveness of the PASTA framework are, to some extent, limited by the quality of pre-trained models, requiring high-quality pre-trained models to achieve optimal performance.
5，Lack of Evaluation of Real-time Performance
The paper does not provide a detailed evaluation of the real-time performance of the PASTA framework, such as processing latency and efficiency in real-time video streams. Real-time performance is critical for many practical applications (such as video surveillance and autonomous driving). The lack of evaluation in this aspect makes it difficult to judge the feasibility and effectiveness of PASTA in real-time environments, potentially limiting its application prospects in scenarios requiring high real-time performance.

---

> ### Author Rebuttal · Authors · 2024-08-05
>
> **W1/Q1 - On the selection of modules**
>
> In our work, we use a Domain Expert to select attributes, a practice that is not far from reality. For example, in fixed-camera scenarios, the mounting perspective is known and whether it will be indoors or outdoors. Lighting can be easily measured with a simple computer vision algorithm, and occupancy can be determined by counting detections. Therefore, the assumption of using a Domain Expert is both reasonable and practical.
>
> **W2/Q2/L2/L3 - On the evaluation on other domains**
>
> We argue that MOT17 and PersonPath22 represent practical, real-world surveillance applications. Unfortunately, due to time constraints, we cannot perform other extensive further evaluations during the rebuttal phase.
>
> **W3/Q4 - On the computational cost**
>
> We compare the GPU memory and time requirements of full fine-tuning versus our approach on the MOTSynth dataset. Our method reduces GPU memory requirements from 13GB to 8.25GB, a reduction of over 35%. This significant decrease is due to the lower number of parameters updated by the optimizer: 42M parameters for standard fine-tuning versus 15M for our approach. Regarding timing, we achieved a reduction from 0.6 seconds per iteration to 0.55 seconds per iteration on average, which is approximately a 10% improvement. Overall training time is reduced from around four days to about 3.5 days on average.
>
> Additionally, our model's adaptability to various scenarios without fine-tuning significantly reduces deployment time across diverse domains.
>
> **W4/Q3 - Synergy between modules**
>
> We kindly refer the reviewer to Tables 5 and 6 and Figure 2 for an analysis of the synergy between different modules. We explore various methods of aggregating and selecting modules. Selecting opposite modules (e.g., poor lighting when the scene has good lighting) results in worse performance than selecting the correct module. Specifically, Figure 2 demonstrates that adding the correct module one at a time monotonically increases both IDF1 and MOTA. We will make sure to highlight these experiments better in the final revision.
>
> **W5 - Module robustness**
>
> The experiments we conducted on MOTSynth, MOT17, and PersonPath22 (Tables 1 to 3) cover a wide range of scenarios, attributes, data volumes, and quality. MOTSynth includes nearly 1.5 million frames and 800 different scenes, PersonPath22 encompasses over 200,000 frames and 236 scenes, while MOT17 comprises 5,316 frames across 14 scenes. These datasets differ significantly in volume and quality, with MOTSynth being a synthetic dataset featuring high-quality images and annotations. PersonPath22 also boasts good image quality and accurate annotations. Conversely, MOT17, having been around longer, presents more challenges due to its relatively lower image quality.
>
> Additionally, our zero-shot experiments demonstrate PASTA's strong generalization capabilities and its resilience to noise, confirming its robustness across diverse conditions and datasets.
>
> **Q5 - Future Work**
>
> The discussion about future work is indeed brief and could be expanded. One aspect that warrants detailed analysis is the routing technique. As suggested by the reviewer, employing reinforcement learning to select the most appropriate modules based on the current scene could be beneficial. Additionally, as suggested by 6tD5, expanding the modules to describe dataset-level characteristics, such as automotive versus pedestrian settings, would be an interesting avenue to explore. Another potential topic for future work involves applying additional domain adaptation techniques, such as GHOST and DARTH (as mentioned in response to oFC1), which could further enhance the adaptability of our approach. We will ensure that these future work discussions are thoroughly addressed in the final version of the manuscript.
>
> **L1 - Increase in complexity**
>
> Our framework introduces a set of parameters for each attribute, which is the only increase in complexity. Training and optimization of modules are conducted in parallel, closely aligning with a standard query-based tracker like MOTRv2. Validation requires only the merging of weights based on the scene to be evaluated. In the deployment phase, there is an additional requirement for selecting or weighting modules.
> This slight increase in complexity offers significant benefits. Specifically, this additional step eliminates the need to fine-tune the model in specific environments. Our approach provides a highly adaptable and efficient tracker by customizing the attributes to suit different scenarios.
>
> **L4 - Dependance on pre-train**
>
> Our framework aims to enhance the knowledge transfer of end-to-end trackers. While the performance of the PASTA framework does depend on pre-trained backbone networks and detectors, we specifically address scenarios where these pre-trained models may perform poorly. Our approach involves fine-tuning on these specific scenarios without altering the pre-trained weights, as we only train a small, disjoint set of parameters. This allows us to retain the benefits of the pre-trained models while improving performance in targeted areas. In our experiments, we compare classical fine-tuning, named MOTRv2-MS, with our modular approach, showing better control and selection of parameters for different scenarios.
>
> **L5 - Real-time performance**
> Our approach does not introduce additional overhead compared to MOTRv2 during inference, so the frames per second (FPS) remain unchanged. Specifically, the YOLOX detector performs at 25 FPS, and MOTRv2 operates at 9.5 FPS with a 2080Ti GPU. When combining these two components, the overall speed is 6.9 FPS.

---

> > ### Author Response · Authors · 2024-08-13
> >
> > Thank you for your detailed and thoughtful review. As we approach the end of the rebuttal period, we wanted to check in to see if there are any further questions or concerns we can clarify. We appreciate your feedback and look forward to your response.

---

### Official Review · Reviewer_6tD5 · 2024-07-11

**Soundness:** 3
**Presentation:** 3
**Contribution:** 3
**Rating:** 6
**Confidence:** 5

**Summary:**

This paper aims to address the domain gap across different multi-object tracking datasets. The proposed method is inspired by LoRA and introduces parameter-efficient fine-tuning for state-of-art end-to-end trackers. Specifically, these trackers are first trained on a large-scale dataset MOTSynth. After that, by shifting several expert attributes and training partial parameters, these trackers can achieve impressive performance on MOT17 and PersonPath22.

**Strengths:**

1. The paper is well-written and easy to follow.
2. The proposed method is novel and interesting. I like it very much!
3. The experiment details are well provided.

**Weaknesses:**

The idea of this paper is attractive for me. However, I still have several minor concerns about the experiments:
1.This paper does not perform zero-shot learning on another popularly used dataset MOT20.
2.The employed attributes in this work, such as lighting, viewpoint, occupancy, location, and camera, seem “weak”. How about the transferring performance on “strong” attributes, like scenes and categories? In other words, how about the zero-shot evaluation performance on other datasets like DanceTrack and KITTI?
3.In Table 1, both model parameters and trainable parameters should be listed for a better comparison. Besides, the table captions in experiments are suggested to provide more descriptions.

**Questions:**

Please refer to the weaknesses.

**Limitations:**

Please refer to the weaknesses.

---

> ### Author Rebuttal · Authors · 2024-08-05
>
> We thank the reviewer for appreciating the novelty and efficacy of our proposed approach. We will answer in the following the doubts of the reviewer:
>
> **W1 - Evaluation on MOT20**
>
> We decided not to evaluate our method on MOT20 since its variety is limited in terms of attributes, as it contains only four similar sequences specifically designed to be crowded. In contrast, the benchmarks we use—MOT17, MOTSynth, and PersonPath22—capture a wide range of attributes and scenarios. The lack of diversity implies that MOT20 as a benchmark might not be representative for assessing our attribute-based approach. For instance, MOT20 lacks scenarios with low or moving cameras. We report in the following the complete MOT20 attribute statistics.
>
> | Per-sequence attributes  | MOT20-TRAIN | MOT20-TEST |
> | ------------------------ | ----------- | ---------- |
> | **Total Sequences**      | **4**       | **4**      |
> | Indoor                   | 2           | 1          |
> | Outdoor                  | 2           | 3          |
> | Camera Low               | 0           | 0          |
> | Camera Mid               | 2           | 3          |
> | Camera High              | 2           | 1          |
> | Moving                   | 0           | 0          |
> | Static                   | 4           | 4          |
> | Bad Light                | 1           | 1          |
> | Good Light               | 3           | 3          |
>
> **W2 - On scenes as attributes**
>
> We chose "weak" (fine-grained) attributes to better align with fine-tuning approaches. However, it would be interesting to explore the addition of "strong" (coarse) attributes to represent completely different scenes or settings. For example, an attribute for automotive settings and another for pedestrian settings. We feel that the rebuttal phase is too short to do justice to such an evaluation, which requires extensive training and experiments. However, we greatly appreciate the reviewer’s suggestion and aim to carry it out in future work.
>
> **W3 - Table 1 parameters**
>
> In Table 1, we reported the total number of trainable parameters (15M for PASTA vs 42M for MOTRv2). Once trained, the total number of parameters in our approach is the same as MOTRv2, i.e., 42M. We will provide a clearer specification of the number of parameters in the final version of the manuscript and will include a better description in the table captions. We thank the reviewer for this suggestion.

---

> > ### Comment · Reviewer_6tD5 · 2024-08-12
> >
> > The authors have addressed the concerns. I keep my rating score.

---

### Official Review · Reviewer_oFC1 · 2024-07-12

**Soundness:** 3
**Presentation:** 4
**Contribution:** 3
**Rating:** 7
**Confidence:** 3

**Summary:**

The paper introduces a fine-tuning framework, denoted PASTA, for multiple-object tracking that is aimed at reducing the cost of tine-tuning large models while mitigating negative inference to improve zero-shot transfer and domain generalization.

During training, the authors independently fine-tune per-domain modules/experts on pre-defined scenario attributes (e.g. lighting, static or moving camera, ...).
Modules only adapt a sub-set of the hyperparameters and the adaptation is done by relying on low-rank adaptation (LoRA). This allows for fine-tuning with a low computational and memory footprint.
More specifically, the authors only adapt batch norm parameters in the backbone and linear layers in the transformer modules.

During inference, the method relies on "expert knowledge" to select the right module for each attribute (e.g. select the "bad lighting" module for the "lighting" attribute). A convex combination of the attribute's modules' weights is then applied (with the selected module having a higher coefficient, defined by a hyper-parameter).

For the zero-shot (transfer) setting, the model is simply set to be a weighted average of all modules.

Experiments demonstrate the effectiveness of the method by comparing performance with the same model trained without the framework. PASTA outperforms the baseline on both the synthetic dataset and on zero-shot transfer (from synthetic to real-world datasets). The authors further demonstrate the reduced forgetting in the ablation studies.

**Strengths:**

- The paper is easy to read, provides a good overview of the methods it relies on (i.e. LoRA and modular deep learning), and includes a good overview illustration.
- The experimental section and the ablation studies demonstrate the effectiveness of the approach and thereby validate the main claims.
- The authors discuss both positive and negative possible societal impacts of the approach.
- "Novelty" is limited as it is mainly an implementation of existing concepts. Nevertheless, the application to a new domain and the good execution make it a useful contribution to the community.

**Weaknesses:**

- The approach is claimed to reduce fine-tuning costs (reduced time and memory footprint), but no time or memory comparison to full tine-tuning is provided.
- The approach is related to domain adaptation and zero-shot transfer, but does not discuss similarities and differences to those approaches in the related work section, and does not compare to such methods in the experimental section.

**Questions:**

- What is the computational gain in terms of time and memory in comparison to the full fine-tuning of MOTRv2?
- How does the method relate to domain adaptation and zero-shot methods?

**Limitations:**

Yes, the authors discuss the potential positive and negative societal impacts of multiple-object tracking, as well as the main limitation of the method (i.e. the reliance on an external "domain expert")

---

> ### Author Rebuttal · Authors · 2024-08-05
>
> As pointed out by the reviewer, a more detailed discussion about the computational costs and the relationship with other domain adaptation methods would enhance the quality of our manuscript. Below, we answer these questions and will integrate this information into the final revision.
>
> **W1/Q1 On the computational cost**
>
> We compare the GPU memory and time requirements of full fine-tuning versus our approach on the MOTSynth dataset.
>
> Our method reduces training GPU memory requirements from 13GB to 8.25GB (for a batch size of 1), a reduction of over 35%. This significant decrease is due to the lower number of parameters updated by the optimizer: 42M parameters for standard fine-tuning versus 15M for our PEFT technique. Regarding timing, we achieved a reduction from 0.6 seconds per iteration to 0.55 seconds per iteration on average, which is approximately a 10% improvement. This is because most of the network parameters remain frozen. Overall training time is reduced from around four days to about 3.5 days on average.
>
> Additionally, since our final model is easily adaptable to a wide variety of scenarios without further fine-tuning, the overall time requirement for deployment across diverse domains is sensibly less compared to standard fine-tuning.
>
> Finally, our approach does not add any overhead during inference, aside from weight merging, which is negligible for stationary attributes, compared to MOTRv2, which maintains a speed of 6.9 FPS on a 2080Ti GPU.
>
> **W2/Q2 Relation with domain adaptation and zero-shot methods**
>
> We acknowledge that the number of works handling domain adaptation and MOT or zero-shot MOT is very limited. We will make a comparison of our approach with the most relatable in the following:
>
> ### **Domain Adaptation methods**
>
> To the best of our knowledge, no tracking-by-query method currently employs domain adaptation. However, we have identified two tracking-by-detection methods that utilize domain adaptation: Simple Cues Lead to a Strong Multi-Object Tracker (GHOST) [30] and DARTH: Holistic Test-time Adaptation for Multiple Object Tracking [29].
>
> **GHOST** is a simple tracker that leverages detections from an off-the-shelf detector. Specifically, GHOST employs "on-the-fly Domain Adaptation," which involves updating the Batch Norm layer statistics during inference (similar to test-time adaptation techniques) in the layers handling reID features. There are several critical differences between GHOST and our approach. First, we only combine already trained modules. Second, the adaptation in GHOST is applied only to the reID module features, whereas our adaptation affects the entire network. Since our approach is end-to-end and query-based, it impacts the tracking and detection components.
>
> **DARTH** employs a test-time adaptation (TTA) technique to adapt the model from a source dataset to a target one using a Knowledge Distillation approach. Each adaptation step requires three forward passes to compute all objective functions, making the process computationally and memory intensive. Additionally, DARTH performs offline TTA, adapting the model using the entire sequence before evaluating it, which is not comparable to our fully online approach. A key difference from our method is that DARTH's requirement of having the entire sequence available for adaptation makes it challenging for real-world use cases. In contrast, our approach only requires simple attributes of the target scene, eliminating the need for further training or adaptation during deployment.
>
> ### **Open-vocabulary methods**
> Recent advancements in zero-shot multiple object tracking demonstrate a significant shift towards open-vocabulary multiple object tracking, a specific setting where textual descriptions define new categories to track. In this context, we recognize the latest works. **OVTrack** is an open-vocabulary tracker based on Faster R-CNN, enhanced with CLIP distillation to learn open-vocabulary capabilities. It also employs a hallucination strategy using denoising diffusion models for robust appearance feature learning. **Z-GMOT** introduces a dataset comprising videos and textual descriptions of target attributes and a tracker that improves detection through advanced grounded language-image pretraining. **OVTracktor** is another open-vocabulary tracker that can detect and segment any category.
>
> Our method, however, does not involve open-vocabulary and language models. Specifically, PASTA focuses on comprehensively transferring domain knowledge of end-to-end trackers with a modular approach. Unfortunately, such open-vocabulary methods are not easily comparable to our approach. They require additional text data during inference and produce results only on open-vocabulary datasets rather than classical MOT benchmarks like MOT17.

---

> ### Comment · Reviewer_oFC1 · 2024-08-12
>
> Thanks to the authors for addressing my concerns by providing more details about the practical gains regarding training time and memory requirements and broadening the related work. I will maintain the "accept" rating.

---

### Official Review · Reviewer_nupV · 2024-07-15

**Soundness:** 3
**Presentation:** 3
**Contribution:** 2
**Rating:** 5
**Confidence:** 5

**Summary:**

The paper presents a multiple object tracking framework that can generalize to new domains by training specialized modules for each scenario attributes.
These modules are trained using Parameter-Efficient Fine-tuning and modular deep learning techniques on a transformer-based tracker.
This tracker is build on the Deformable DETR framework with a ResNet backbone for image feature extraction.
Extensive experiments are performed on both synthetic and real datasets to compare with some recent track-by-detection and track-by-attention approaches.
It shows that the proposed approach can generalize well on unseen datasets and reduce negative interference during training.

**Strengths:**

1. The proposed modular architecture is significant and origin in solving MOT problem.
2. Experiments support and prove the proposed architecture's ability to generalize on new domains, i.e. real dataset from synthetic dataset.

**Weaknesses:**

1. The authors only conduct a synth-to-real zero-shot experiment. It would be better to see real-to-real experiment where one can train on one dataset and test on a new dataset.
2. It would be great to have some statistic on the training dataset in terms of the attributes, e.g. how many frames / videos have high/low occupancy, etc.
3. One missing aspect in the ablation studies is how the task vector is incorporate to the pre-trained tracker contribute to the final results. For examples, incorporating on the backbone, encoder or decoder only and the whole network can have different results.
4. The proposed approach seems to be limited by the availibility of the attribute data for training each scenario-specific module.

**Questions:**

1. Is it necessary to have training data for all possible cases to train all the parameter modules?
2. How does the model trained on the MOT17 training set perform on the PersonPath22 test set? This will also demonstrate the generalization ability of the model in the zero-shot setting on a new domain.
3. It is not clear why MOTRv2 is not compared in the fully-trained section in Tables 2 and 3 while TrackFormer [24] is included in both fully-trained and zero-shot.

**Limitations:**

Yes. The authors provided some analysis on limitations and social impacts.

---

> ### Author Rebuttal · Authors · 2024-08-05
>
> We thank the reviewer for the valuable questions and for having appreciated the originality and efficacy of the proposed approach.
>
> **W2 - Dataset statistics**
>
> We report the requested details in the following tables, which provide statistics on the employed datasets divided by per-sequence and per-frame attributes
>
> | Per-sequence attributes | PP22    | MOT17   | MOTSYNTH |
> | -- | ------- | ------- | -------- |
> | **Total Sequences**     | **236** | **12**  | **764**  |
> | Indoor| 61      | 2       | 57       |
> | Outdoor| 175     | 12      | 707      |
> | Camera Low| 162     | 10      | 479      |
> | Camera Mid| 68      | 4       | 244      |
> | Camera High| 6       | 0       | 41       |
> | Moving| 60      | 8       | 220      |
> | Static| 176     | 6       | 544      |
> | Bad Light| 23      | 0       | 189      |
> | Good Light| 213     | 14      | 575      |
>
>
>
> | Per-frame attributes | PP22   | MOT17 | MOTSYNTH |
> | -- | ------ | ----- | -------- |
> | **Total Frames**     | **203653** | **5316**  | **1375200**  |
> | Occupancy Low        | 44%    | 24%   | 29%      |
> | Occupancy Mid        | 53%    | 54%   | 68%      |
> | Occupancy High       | 3%     | 22%   | 3%       |
>
> **W3 - Ablation of task vectors contribution**
>
> We agree that an additional ablation study on where to apply the task vectors might clarify which components are most critical for adaptation over scenario attributes. To investigate this aspect, we trained three versions of PASTA by omitting the task vectors from the encoder, decoder, or backbone, respectively. We will report these results computed on MOTSynth in the final manuscript:
>
> || HOTA | IDF1 | MOTA | DetA | AssA |
> |----|------|------|------|------|------|
> | MOTRv2-MS| 52.4 | 56.5 | 61.9 | 56.4 | 49.0 |
> | PASTA - no decoder| 51.5 | 56.0 | 58.9 | 53.6 | 49.8 |
> | PASTA - no encoder| 52.4 | 56.9 | 61.2 | 55.7 | 49.7 |
> | PASTA - no backbone| 52.5 | 57.0 | 61.5 | 55.6 | 49.9 |
> | PASTA - all| 53.0 | 57.6 | 62.0 | 56.2 | 50.4 |
>
> The results indicate that not applying task vectors to the decoder significantly degrades detection and association metrics. We believe that this degradation can be explained by considering the crucial role of the decoder. The decoder must indeed gather information from detection, tracking, and proposal queries while simultaneously integrating visual information from the encoder. Consequently, not adapting the decoder prevents the architecture from effectively leveraging queries and visual cues. The encoder also contributes substantially, though to a lesser extent than the decoder, as it primarily refines and contextualizes visual features from the backbone. The backbone shows the smallest contribution.
>
> **Q1/W4 - Attributes availability during training**
>
> During training, it is important to identify the relevant attributes for the specific problem at hand. In our approach, the selection of attributes was primarily guided by common sense and our experience with the task. Also, we chose attributes that are generally applicable to most use cases. However, this selection serves as a proof of concept. In practical applications, the selection could be further refined to reflect specific characteristics of the problem under consideration.
>
> For instance, in a surveillance scenario characterized by indoor cameras, the indoor/outdoor attribute may become unnecessary. Conversely, an attribute for weather conditions might be valuable in locations with variable weather.
>
> The need for predefined attributes is easily addressable using automatic or semi-automatic classifiers. For example, an analysis of the brightness level can easily classify lighting conditions or a straightforward detector can be exploited to count objects of interest in the scene, thereby classifying crowd density.
>
> **W1/Q2 - Real-to-real (MOT17 → PersonPath22)**
>
> We report below the results of the model trained on MOT17 and evaluated in zero-shot on PersonPath22.
>
> | MOT17->PP22 | HOTA | IDF1 | MOTA | FP |   FN   | IDsw |
> |-------------|------|------|------|----|--------|------|
> | MOTRv2      | 43.9 | 51.5 | 48.6 |8304| 119391 | 5342 |
> | PASTA       | 46.1 | 54.6 | 50.8 |7895| 114620 | 4702 |
>
> To provide a comparison, we train MOTRv2 on MOT17 and test their performance on PersonPath22. Our approach yields better results compared to fine-tuned MOTRv2, demonstrating that exploiting modules improves the model's generalization capabilities on new and/or real domains. We will include these findings in the final revision of the manuscript.
>
> **Q3 - Missing MOTRv2 full tuning in Tables 2 - 3**
>
> We agree that including fine-tuned MOTRv2 results would provide a better understanding of the zero-shot experiments. We primarily did not undertake it due to time constraints and computational overheads. Herein, we remedy this and report the results for Table 2 (MOT17):
>
> || HOTA | IDF1 | MOTA | DETA | ASSA |
> | ------------------------- | ---- | ---- | ---- | ---- | ---- |
> | MOTRv2 (Fine-tuned by us) | 66.8 | 78.9 | 73.2 | 62.5 | 71.4 |
> | MOTRv2-MS (zero-shot)     | 62.6 | 73.0 | 67.6 | 60.3 | 65.5 |
> | PASTA (zero-shot)         | 64.0 | 74.9 | 68.1 | 60.4 | 68.3 |
>
> Notably, our zero-shot approach performs similarly to its fine-tuned counterpart. We appreciate the reviewer highlighting this missing comparison, which strengthens our findings.
>
> Regarding Table 3 (PersonPath22), we have included fully trained methods on PersonPath22 as reported in the PersonPath22 paper. We could not train MOTRv2 on this dataset primarily because the authors of the PersonPath22 did not release the YOLOX weights they used for the other methods listed in their table nor provide interpolation scripts for the training-set annotations. Since training MOTRv2 on PersonPath22 is not crucial for the PASTA evaluation, and given the abovementioned limitations, we opted to refrain from reporting the full tuning of MOTRv2.

---

> > ### Comment · Reviewer_nupV · 2024-08-12
> >
> > The authors have adequately addressed the concerns raised. I appreciate their efforts and will maintain my rating.

---

### Author Rebuttal · Authors · 2024-08-05

We thank all the reviewers for their thorough evaluations and constructive feedback. These comments and suggestions have contributed to enhancing the overall quality of our work. Below, we summarize the strengths and weaknesses reported.

**Strengths:** We greatly appreciate the recognition of the **novelty** of our work by reviewers nupV, 6tD5, 4fn5, and Vx5C, as well as the **validation** of our experimental section by all reviewers. Reviewers oFC1, 6tD5, 4fn5, and Vx5C appreciated the **clarity and readability** of our writing and the effectiveness of the overview figure. Additionally, reviewers nupV, oFC1, and 4fn5 acknowledged the contribution of our work. For instance, 4fn5 remarked, *"PASTA performs excellently in handling heterogeneous scenarios, demonstrating significant innovation and research value."*

**Weaknesses:** Reviewers nupV, oFC1, and Vx5C highlighted the need for further comparisons with zero-shot or domain adaptation methods. Additionally, there was a common concern about the lack of a detailed computational analysis (4fn5, Vx5C). To address these weaknesses, we compared our methods with related zero-shot and domain adaptation MOT methods (oFC1) and evaluated other by-detection methods in a zero-shot setting (Vx5C). We also elaborated on the choice of attributes (nupV, 6tD5, 4fn5, Vx5C).
- nupV major concern was related to a missing experiment and ablations and required additional clarity on the attribute availability. We report the real-to-real zero-shot evaluation on PersonPath22 and further ablations as requested. Additionally, we clarify the doubts on the training section.
- oFC1 noted the lack of quantitative computational analysis and a study of the relationship with similar works. We report the time and memory requirements of our approach and its fine-tuned counterpart. Furthermore, we provide an analysis of related domain adaptation and zero-shot works.
- 6tD5 had minor concerns regarding the lack of experiments on MOT20 and the performance on “strong” attributes. We provide a motivation for this lack and the attribute choice.
- 4fn5 suggested several clarity improvements, which we will integrate in the final revision of the paper. Their major concern is related to the choice of the attributes and the computational analysis. We provide a discussion on the attribute selection and a quantitative computational analysis.
- Vx5C highlighted the need for a zero-shot evaluation of ByteTrack and other tracking-by-detection methods. We report this comparison on PersonPath22 and a threshold analysis on MOTSynth. We also provide an explanation of the attribute choice and an analysis of performance without LoRA.

We address each reviewer's concerns and recommendations in detail within our individual responses to each review. We will ensure that these improvements will be incorporated into the final revision of our manuscript.

---

### Decision · Program_Chairs · 2024-09-25

**Decision:**

Accept (poster)

**Comment:**

The rebuttal provided clarifications about the proposed method and its analysis that were useful for assessing the paper's contribution and responded adequately to most reviewer concerns. All reviewers recommend acceptance after discussion (with three borderline accepts, one weak accept and one accept), and the ACs concur. The final version should include all reviewer comments, suggestions, and additional clarifications from the rebuttal.